# *Borrelia miyamotoi*: A Comprehensive Review

**DOI:** 10.3390/pathogens12020267

**Published:** 2023-02-07

**Authors:** Dawn W. Cleveland, Cassidy C. Anderson, Catherine A. Brissette

**Affiliations:** Department of Biomedical Sciences, University of North Dakota, Grand Forks, ND 58202, USA

**Keywords:** *Borrelia miyamotoi* disease (BMD), *Ixodes*, Lyme disease, relapsing fever, reservoir species, tick-borne disease, vector

## Abstract

*Borrelia miyamotoi* is an emerging tick-borne pathogen in the Northern Hemisphere and is the causative agent of *Borrelia miyamotoi* disease (BMD). *Borrelia miyamotoi* is vectored by the same hard-bodied ticks as Lyme disease *Borrelia*, yet phylogenetically groups with relapsing fever *Borrelia,* and thus, has been uniquely labeled a hard tick-borne relapsing fever *Borrelia*. Burgeoning research has uncovered new aspects of *B. miyamotoi* in human patients, nature, and the lab. Of particular interest are novel findings on disease pathology, prevalence, diagnostic methods, ecological maintenance, transmission, and genetic characteristics. Herein, we review recent literature on *B. miyamotoi*, discuss how findings adapt to current *Borrelia* doctrines, and briefly consider what remains unknown about *B. miyamotoi*.

## 1. Introduction

The genus *Borrelia* contains Gram-negative, obligate extracellular parasitic bacteria with a unique spiral morphology. Known as spirochetes, *Borrelia* owe their shape and spiraling motility to their periplasmic flagella [1]. *Borrelia* can be divided into two well-defined groups: the Lyme disease (LD) group, vectored by hard ticks, and the relapsing fever (RF) group, vectored by soft ticks and lice. Classically, these groups were highly distinct with separate vector genera, unique transmission dynamics, and differing disease outcomes [2,3]. Yet, the discovery of *Borrelia miyamotoi*, phylogenetically grouped with RF spirochetes, but vectored by the same hard ticks as LD spirochetes, disturbed this archetype and fueled research [4].

In this review, we discuss novel *B. miyamotoi* research in recent publications and examine how new data fit into existing knowledge.

## 2. Background

Ticks are capable of vectoring numerous pathogens, with the most prevalent vector-borne illness in the Northern Hemisphere being Lyme disease. Also known as Lyme borreliosis, LD is the result of infection with species in the *Borrelia burgdorferi* sensu lato (*Bb*sl) complex, which are vectored by hard-bodied *Ixodes* ticks. The primary species responsible for human LD are *B. burgdorferi*, *B. afzelii*, and *B. garinii* [2]. Lyme disease is commonly characterized by the formation of an erythema migrans skin lesion, often at the bite location, in conjunction with flu-like symptoms of headache, fatigue, muscle aches, and fever along with joint pain and swelling. *Bb*sl spirochetes can disseminate beyond the bite site to additional tissues. This colonization can result in serious clinical manifestations of the joints, heart, and central nervous system (CNS), known as Lyme arthritis, carditis, and neuroborreliosis, respectively [5,6,7,8]. The treatment for LD is an antibiotic regimen; however, symptoms can persist, resulting in post-treatment Lyme disease syndrome [5,7].

Another borreliosis found in the Northern Hemisphere, Africa, and Central America is relapsing fever. Relapsing fever is the result of infection by several species of *Borrelia* vectored primarily by soft-bodied argasid ticks, the exception being *B. recurrentis* vectored by lice. Clinical presentations of RF typically consist of a high fever for a few days, followed by an approximately week-long period of well-being, followed by another relapse. Relapses can repeat multiple times without antibiotic treatment. Additional RF symptoms are headache, body aches, and abdominal pain. Historically, outbreaks are caused by louse-borne *B. recurrentis* and occur during wartime, periods of poor hygienic conditions, and refugee crowding [3].

*Borrelia miyamotoi*, a new tick-borne *Borrelia* species, was discovered in Japan in 1995 when spirochetes were isolated from *Ixodes persulcatus* midguts and *Apodemus argenteus* blood [4]. Despite being vectored by hard ticks, *B. miyamotoi* was found to be genetically distinct from the *Bb*sl complex and was more closely related to RF species. At the time, there were no cases of RF in the region; this led researchers to speculate that this new *Borrelia* species was a tick endosymbiont maintained by an enzootic transmission cycle [4].

The pathogenicity of *B. miyamotoi* was realized in Russia in 2011 when 51 patients with suspected tick bites presented with a nonspecific febrile illness. The patients were found to be infected with *B. miyamotoi* by *B. miyamotoi*-specific polymerase chain reaction (PCR) of the blood or anti-borreliae immunoglobulins in the serum. The patients’ symptoms were somewhat dissimilar to RF with fever, headache, chills, fatigue, and myalgia. One patient had a relapse, which may have been prevented in other patients due to antibiotic treatment [9]. The RF-like illness caused by *B. miyamotoi*, now known as *B. miyamotoi* disease (BMD), has since been widely diagnosed and is considered an emerging public health threat [10].

## 3. Disease

Human infection with *B. miyamotoi* can result in BMD. Symptoms vary depending on the constitution of the patient. Immunocompetent, and otherwise healthy, patients present with milder, flu-like symptoms: fever, fatigue, sleepiness, chills, muscle and joint stiffness, aches and pains, and nausea [11,12,13]. While uncommon, relapses of febrile episodes can occur [11]. It is possible that many patients with BMD do not seek medical attention due to short-lived and mild symptoms. Similar to the mechanism of relapses in other RF species, *B. miyamotoi* exhibits antigenic variation, allowing spirochetes to evade adaptive immune responses [14,15,16,17]. In the blood, *B. miyamotoi* has numerous mechanisms to evade complement-mediated killing, allowing for the rapid growth and multiplication of spirochetes that result in the symptoms of BMD [11,14,15,16].

The adaptive immune response is important for controlling *B. miyamotoi* infection [15]. Studies of RF species *B. hermsii* show that infection clearance is antibody mediated [16]. *Borrelia miyamotoi* activates dendritic cells which phagocytose the bacteria and produce interleukins IL-8, IL-6, and IL-12, as well as tumor necrosis factor alpha (TNF-α) [18]. These cytokines further stimulate and signal immune activation for inflammation, recruitment of additional immune cells, infection clearance, host-defense, and T-cell differentiation [18]. Immunocompetent mouse models of BMD only develop transient infections, whereas immunodeficient mice mouse models develop persistent infections [19]. This suggests that healthy patients infected with *B. miyamotoi* may be able to clear the infection without medical intervention.

*Borrelia miyamotoi* infection in immunocompromised patients can be much more severe [20]. In combination with generalized flu-like symptoms, immunocompromised patients often exhibit reduced cognition, disturbed gait, memory deficits, confusion, and other neurological deficiencies resultant of meningoencephalitis [21]. Additionally, hearing loss, weight loss, uveitis, iritis, neck stiffness, and photophobia have been reported [20]. The designation “immunocompromised” is broad; however, hospitalization reports of severe BDM are frequently seen in patients prescribed B-cell depletion therapies, such as rituximab, other cancer immunotherapeutics, or immunosuppressants for rheumatoid arthritis [20,21,22,23,24,25]. While these medications are often seen in conjunction with meningoencephalitis and other serious symptoms, there is no finite list of medications, treatments, or immunodeficiencies that can give rise to severe BMD.

In patients lacking a complete immune response, the austerity of symptoms is possibly due to unchecked growth of *B. miyamotoi* in the blood. Similar to *B. hermsii*, it can be speculated that the humoral immune response is necessary to control *B. miyamotoi* infection [16]. This suggests that defects in antibody production allow spirochetes to disseminate from the blood and colonize farther tissues, resulting in severe BMD symptoms. It can be inferred that BMD-associated meningoencephalitis is the result of *B. miyamotoi* colonizing the CNS, as microscopic analysis of cerebrospinal fluid (CSF) collected from BMD patients with meningoencephalitis shows visible spirochetes [21,22,23]. However, the dynamics of host–pathogen interactions leading to inflammation are unknown.

There are no official clinical guidelines for the treatment of diagnosed or suspected BMD. Thereby, treatment falls to the guidelines for LD and RF: an antibiotic regimen, commonly doxycycline and ceftriaxone. Ampicillin, azithromycin, and vancomycin, or a combination thereof, have also been used to treat BMD. Antibiotic treatment engenders a full recovery with rare cases of persistent fatigue [20,26]. A systematic review and meta-analysis by Hoornstra et al. affirmed that doxycycline is the preferred treatment for adults presenting with no neurological complications, and ceftriaxone is the preferred treatment for adults with BMD-associated meningoencephalitis [26]. Research has confirmed in vitro that both clinical and tick isolates of *B. miyamotoi* are susceptible to doxycycline, ceftriaxone, and azithromycin, but show resistance to amoxicillin [27]. Albeit rarely, Jarisch-Herxheimer reactions have occurred with antibiotic treatment of BMD [22].

## 4. Diagnosis

Diagnosis of BMD and *B. miyamotoi* infection is possible using microscopy, PCR, and serodiagnosis, either as single tests or in combination, with the latter two methods being the most common [11]. However, there is no official standardized diagnostic technique [28].

### 4.1. Microscopy

Spirochetes can be visualized in blood and CSF using dark-field microscopy, or with Giemsa staining or acridine orange staining [21,22,23]. Microscopy can confirm a *Borrelia* species infection; however, it cannot be used to distinguish between species and has low sensitivity [29]. In order to see spirochetes in a blood smear or CSF sample, the density needs to be greater than 10^4^ cells/mL [11,26]. During the acute phase of BMD, usually within a few days of symptom onset, spirochete density in the blood ranges from 10^1^–10^5^ cells/mL [30]. The presence of *B. miyamotoi* in the CSF is poorly understood. Thus, there is no guarantee that spirochete density will be high enough for visualization.

*Borrelia miyamotoi* can be cultured from a patient’s whole blood, serum, or CSF into specialized media, and organisms can then be visualized. However, as spirochetes may take longer than two weeks to grow to a visible density and culturing *Borrelia* species is notoriously tricky, it is neither timely nor practical to use culturing for diagnosis [31].

The visualization of spirochetes in the blood or CSF confirms *Borrelia* species infection, but a lack of visible spirochetes does not rule out infection. Microscopy can support a BMD diagnosis but should be used in combination with other methods.

### 4.2. PCR

Diagnosis of BMD often uses PCR assays performed on whole blood, serum, or CSF and typically targets *glpQ* and/or *flaB* [11,22]. It is best to use PCR to diagnose BMD when spirochetes are most likely to be detectable during symptom presentation and prior to antibiotic treatment [20,30]. Human borrelioses all have detectable periods of spirochetemia, albeit brief for *Bb*sl, and overlapping geographical regions [32,33,34]. Thus, a PCR assay must be sensitive enough to detect *B. miyamotoi* and specific enough to distinguish between *B. miyamotoi*, *Bb*sl, and RF species.

A clinically relevant assay was developed to detect and differentiate between *B. miyamotoi*, louse-borne RF, and soft tick-borne RF species. The semi-multiplex real-time PCR assay is highly sensitive with a lower detection limit of ten genome copies and uses a nested approach to first broadly amplify and detect RF species and *B. miyamotoi*. A series of probes then differentiate between groups [35]. If implemented in a clinical setting, this assay would readily test for RF and BMD in acute phase patients.

### 4.3. Serodiagnosis

*Borrelia miyamotoi* infection results in the production of immunoglobulins to spirochete proteins, and serodiagnosis utilizes the presence of these reactive antibodies to diagnose a past or present infection. The quantity and variety of antibodies produced fluctuates depending on the route of inoculation, whether via tick bite or needle stick, if antigenic variation occurs, and the duration and dissemination of the infection [14,15,16,36,37]. Immunoglobulins IgM and IgG are produced in response to *B. miyamotoi* infection, with IgM peaking around one week post infection and IgG peaking around three weeks post infection. While IgG is highly specific for the antigen against which it is produced, IgM is not and is more prone to higher background and cross-reactivity [38,39].

Timing is important when utilizing serology for a diagnosis. At the early onset of BMD symptoms, IgM may be present at low levels, while IgG will not be present [38,39]. Absent or undetectable antibodies could result in a false negative test. Microscopy paired with PCR is more suited for acute diagnosis [20,21,22,23,30]. Patients presenting with chronic BMD symptoms multiple weeks after disease onset, or for retrospective studies, are better suited to diagnosis via serology. The indefatigable nature of anti-*B. miyamotoi* antibodies is not fully understood; however, the persistence of *Borrelia*-specific antibodies is correlated to the duration and dissemination of the infection [36]. As antibody production is integral for serodiagnosis, the ability to diagnose immunocompromised patients via serology may be limited due to complications surrounding reduced humoral responses to *B. miyamotoi*.

Glycerophosphodiester phosphodiesterase (GlpQ) is widely used for serodiagnosis, but its reliability as a serological indicator of BMD has recently been called into question [13,15,40,41]. For PCR, *glpQ* is reliable as it directly detects the bacteria. However, as serodiagnosis indirectly detects infection by measuring reactive antibodies to a particular antigen or antigens, cross-reactivity can potentially occur between similar proteins from different pathogens [42].

Anti-*B. miyamotoi* antibodies can be detected using conventional assay systems such as an ELISA or a Western blot [43,44]. Ideally, the best candidate antigens for serodiagnosis should be highly conserved and detectable in both early and late infection with low cross-reactivity to other tick-borne and common bacterial pathogens. Reactivity to GlpQ is often used in combination with flagellin and p66 proteins; however, neither of these are species specific [20]. Despite widespread use for serodiagnosis, GlpQ has been shown to be neither highly sensitive nor specific [41]. Fortunately, researchers have strove to uncover methods to improve GlpQ accuracy and identify additional reliable markers.

Assaying for GlpQ reactivity in conjunction with variable major proteins (Vmps), particularly variable large protein (Vlp)-15/16, improves the sensitivity and specificity of *B. miyamotoi* detection [37]. This coalition leans heavily on cooperation as Vmps alone are prone to cross-react with orthologous *Bb*sl protein Vmp-like sequence expressed (VlsE) [45,46]. This dual-marker technique using both GlpQ and Vlp-15/16 was found to be an excellent diagnostic option as IgM against GlpQ and Vmps peaks between 11 and 20 days and IgG peaks between 21 and 50 days. This combination increased specificity to 100% and sensitivity to 79% on days 11–20, and increased specificity to 98.3% and sensitivity to 86.7% on days 21–50 [37]. Thus, testing for seroreactivity to both Vmps and GlpQ can improve BMD diagnostic accuracy.

Whereas GlpQ and Vlp-15/16 test best in combination, *Borrelia miyamotoi* membrane antigen (BmaA) is a solitary candidate for serodiagnosis. This protein is an externally located cellular putative lipoprotein of *B. miyamotoi*, and anti-BmaA antibodies can be detected in the serum of confirmed BMD patients between 38 and 100 days after disease onset [47]. Results after 100 days were inconclusive and additional testing is necessary. BmaA has little to no observable reactivity in LD patients or with RF species *B. turicate*, indicating that BmaA may be species specific [47]. Beyond BmaA, more than 400 immunoreactive peptides, most of which mapped to Vmps, were recently identified as potential targets for future serology studies [48]. Hence, while GlpQ may not be ideal for serodiagnosis, there are multiple other options.

## 5. Human Cases and Infection Prevalence

The National Institute of Allergy and Infectious Diseases classifies *B. miyamotoi* as an emerging infectious pathogen and BMD as an emerging infectious disease [10]. Even as *B. miyamotoi* becomes more prominent, the ambiguity and often self-limiting nature of BMD symptoms suggest that many cases are likely subclinical. Patients who do seek medical treatment may be misdiagnosed with other tick-borne diseases (TBD), particularly LD. This mistaken serodiagnosis of BMD as LD can be due to the C6 peptide of the *Bb*sl VlsE protein. Reactivity to the C6 peptide is widely used for LD serodiagnosis, and infection with *B. miyamotoi* leads to cross-reactive antibodies, which may result in a false-positive LD test [49]. As antibiotic treatment for LD is successful in treating BMD, an accurate diagnosis of BMD may never be reached. All of this, combined with low awareness among physicians and the general public, indicates that the known prevalence of BMD may not be wholly accurate. The most updated and thorough conglomeration of reported diagnoses of BMD and retrospective studies of banked human sera reactivity is by Hoornstra et al. in their systematic review and meta-analysis [26].

Including the first cases of BMD diagnosed in Russia, there have been 561 total diagnoses: 367 in Russia, 101 in the United States, 57 in France, 30 elsewhere in Asia, and 6 elsewhere in Europe [26,50,51,52,53,54]. The higher number of cases in Russia may be due to increased awareness of *B. miyamotoi* prompting more testing, increased transmission abilities of *I. persulcatus* ticks, or greater virulence of Russian strains, leading to more hospitalizations. As there are no official diagnostic guidelines for BMD, criteria for BMD diagnosis may vary between institutions.

The reactivity of banked sera to *B. miyamotoi* recombinant proteins, primarily GlpQ and FlaB, can be measured using ELISA and Western blots, either singly or in conjunction as a two-tier test. Across these retrospective studies, 45,608 samples were tested from North America, Europe, and Asia and separated into varying categories depending on risk, confirmed infection, and cross-reactivity. Seroprevalence was 4.6% in the high risk for *B. miyamotoi* infection group, 4.8% in the suspected LD group, 11.9% in the suspected TBD group, and 1.3% in healthy controls [26]. Prevalence across all groups could indicate that *B. miyamotoi* infection is far more common than previously thought, detection methods are prone to false positives and cross-reactivity, or coinfections with other tick-borne pathogens (TBP) produce convoluted results [11].

## 6. Ticks

The geographic distribution of *B. miyamotoi* in ticks is far greater than that of BMD in humans. This may be an insight on transmission capabilities, or it may be that more surveys have been conducted on ticks. *Borrelia miyamotoi* is vectored by *I. scapularis* in the northeastern and northcentral United States and Canada, *I. pacificus* in the western United States, *I. ricinus* in Europe, *I. persulcatus* in Europe and Asia, and *I. ovatus, I. nipponensis* and *I. pavoloskyi* in Asia [26,50,55]. *Borrelia miyamotoi* has been detected in all known vectors of *Bb*sl [43]. There are reports of *B. miyamotoi* in other tick species, including *Dermacentor reticulatus* in Russia and *Haemaphysalis* species in China and Slovakia [56,57,58,59]. These species are not competent vectors for *Bb*sl; however, it is unclear if they are capable of successfully vectoring *B. miyamotoi* [60,61]. Vector competence is the ability of a tick to serve as a disease vector. Vector competence is influenced by environmental factors, such as vector density, longevity, and fitness, and genetic factors, such as host preference and duration of attachment [62]. Table 1 has an updated list of countries with reports of ticks infected with *B. miyamotoi.* The systematic review and meta-analysis by Hoornstra et al. analyzed *B. miyamotoi* prevalence in ticks, which is outlined in Table 2. The overall prevalence of *B. miyamotoi* in questing *Ixodes* ticks is 1.1%, but varies by region and dominant tick species in that region [26].

A survey of *Ixodes* ticks in the United States from 2013 to 2019 screened nine *Ixodes* species, but only detected *B. miyamotoi* in *I. pacificus* (14/1497, 0.94%) and *I. scapularis* (594/34,621, 1.72%). The average infection prevalence in nymphs and adults was similar; there was no mention of larval infection. *Borrelia miyamotoi* was found in ticks in 19 states, with infection prevalence being 0.5–3.2%. Ticks tested in 20 additional states were all negative [63]. *Borrelia miyamotoi* has been detected in all Canadian provinces, excluding Newfoundland, in *I. scapularis* ticks [34]. There are certain states and provinces in which *B. miyamotoi* infection has not been reported, which may be due to a paucity of competent vector species in that region, spirochetes present but below detection limits, or simply that *B. miyamotoi* is not in that region.

Natural coinfections in *Ixodes* ticks occur, as these arthropods vector a number of TBP beyond *Borrelia* species [33]. A survey of *Ixodes* ticks in the United States found over half of all *B. miyamotoi*-infected *I. scapularis* ticks had concurrent infections (351/594, 59.09%). Half of all *B. miyamotoi*-infected ticks had a dual infection (293/594, 49.33%) with either *B. burgdorferi* (220/594, 37.04%), *Anaplasma phagocytophilum* (43/594, 7.24%), or *Babesia microti* (30/594, 5.05%). Additionally, 52 ticks (8.75%) had triple infections and six ticks (1.01%) had quadruple infections. No concurrent *B. miyamotoi* infections were seen with *B. mayonii* or *Ehrlichia muris*-like agent [33]. This could be due to coincidence, the result of geographic constraints, and/or competitive habitation in the vector ticks. Regardless, these data indicate that concurrent infections in ticks are common, and physicians should be aware that tick bite patients may be exposed to multiple TBPs.

In the tick, *B. miyamotoi* disseminates throughout the body into the acini, salivary gland ducts, basal lamina of the midgut, epithelium of the Malpighian tubes, female ovarian tissues, male testes, CNS, and near the mouthparts [64]. During tick colonization, *B. burgdorferi* species have outer surface membrane proteins that adhere to receptors in the midgut lumen and protect spirochetes from the ingested host blood containing complement and antibodies [62,65]. It is unknown if *B. miyamotoi* uses similar mechanisms for tick colonization as *Bb*sl.

Additionally, ticks infected with *B. miyamotoi* are found to have significantly higher numbers of *Borrelia* cells than those infected with *Bb*sl [66,67]. This is a proposed evolutionary trait to compensate for the lower prevalence of *B. miyamotoi* in tick populations (~1%), compared with *Bb*sl (~12%) [26,67,68]. Furthermore, *B. miyamotoi* reach higher levels in the blood than *Bb*sl species, and thus, ticks ingest more bacteria per blood meal [69]. Other *Borrelia* species, including *Bb*sl and *B. hermsii*, have been shown to replicate within ticks; however, it is not known if *B. miyamotoi* replicates within ticks [70,71].

When transmitted from tick to host, *Bb*sl uses outer surface protein C (OspC) to bind tick salivary protein Salp15, which provides protection from the mammalian immune response [72]. *Borrelia hermsii* variable tick proteins (Vtp) are homologous to OspC and convey a similar function [73]. The variable small proteins (Vsp) in *B. miyamotoi* are homologous to OspC and Vtp and likely aid in transmission from tick to vertebrate host [14]. There is still much to learn about the intricacies of *B. miyamotoi* tick colonization and transmission.

**Table 1 pathogens-12-00267-t001:** Countries with *B. miyamotoi*-infected ticks.

Country	Tick
Austria	*I. ricinus* [74]
Belarus	*I. ricinus* [75]
Belgium	*I. ricinus* [76,77]
Canada	*I. scapularis* [78]
China	*I. persulcatus* [79], *H. longicornis* [57], *H. concinna* [58]
Czechia	*I. ricinus* [80]
Denmark	*I. ricinus* [81]
Estonia	*I. ricinus, I. persulcatus* [82]
Finland	*I. ricinus* [83,84,85]
France	*I. ricinus* [81,86,87]
Germany	*I. ricinus* [76,88]
Ireland	*I. ricinus* [89]
Italy	*I. ricinus* [76]
Japan	*I. persulcatus* [4], *I. ovatus* [90]
Moldova	*I. ricinus* [91]
Mongolia	*I. persulcatus* [90,92]
Netherlands	*I. ricinus* [76,81,93,94]
Norway	*I. ricinus* [95]
Poland	*I. ricinus* [96,97]
Portugal	*I. ricinus* [98]
Russia	*I. persulcatus* [99]
Serbia	*I. ricinus* [100]
Slovakia	*I. ricinus* [59,101], *H. inermis* [59]
South Korea	*I. nipponensis* [55]
Spain	*I. ricinus* [102]
Sweden	*I. ricinus* [67,76]
Switzerland	*I. ricinus* [103]
Turkey	*I. ricinus* [104]
Ukraine	*I. ricinus* [105]
United Kingdom	*I. ricinus* [76]
United States	*I. pacificus* [106], *I. scapularis* [107]

**Table 2 pathogens-12-00267-t002:** Prevalence of *B. miyamotoi* in ticks ^a^.

	Questing Tick Prevalence	*B. miyamotoi* Prevalence
*I. scapularis*	28.0%	1.1%
*I. pacificus*	14.8%	0.7%
*I. ricinus*	52.2%	1.0%
*I. persulcatus*	5.0%	2.8%

The combined surveys listed in Hoornstra et al., 2022 covered a total of 165,637 questing ticks. Tick species prevalence is calculated as a proportion of total questing ticks surveyed. *B. miyamotoi* prevalence was calculated as proportion of the tick species. ^a^ adapted from Hoornstra et al., 2022 [26] under creative commons license CC BY 4.0.

## 7. Animal Infections

The majority of animals infected with *B. miyamotoi* are small mammals, particularly rodents, with a minority of infections being in larger mammals and birds [108]. An important distinction needs to be made between animal infections, where the animal tissue tested positive for *B. miyamotoi* (Table 3), and animal-associated tick infections, where the animal tissue was not tested, but the tick(s) attached to the animal tested positive for *B. miyamotoi* (Table 4). *Borrelia miyamotoi* has also been detected in both birds and bird-associated ticks; however, there has been minimal follow-up research to interpret the role of birds in *B. miyamotoi* perpetuation and maintenance (Table 3 and Table 4) [94,109,110].

The multiple reservoir species identified for *B. miyamotoi*, all of which are rodents, are *Peromyscus*, *Apodemus*, and *Myodes* species [69,108,111,112]. Reservoir rodent species have been identified carrying *B. miyamotoi* in both rural and urban environments, posing an increased public health risk [83,84,85]. A notable potential new reservoir species in North America is the jumping mouse, *Napaeozapus insignis*. In a survey of *B. miyamotoi* in Atlantic-Canadian wildlife, *N. insignis* had a much higher infection prevalence than other known reservoir species tested in the same survey. Unfortunately, the role of the jumping mouse in *B. miyamotoi* transmission is unknown, as specimens were collected via public participation and transmission studies cannot be conducted post-mortem [113].

It has been speculated that white-tailed deer may be a reservoir host for *B. miyamotoi,* as a study found that deer-associated ticks had higher infection prevalence than questing ticks [114]. Unlike *B. burgdorferi* that is lysed by deer blood complement upon tick ingestion, it appears that *B. miyamotoi* remains viable in the tick upon ingestion of deer blood. A study found that engorged female *I. scapularis* ticks collected from white-tailed deer produced infected offspring [114,115,116]. Reservoir competence has not been characterized in white-tailed deer.

Reservoir species must be able to be infected with *B. miyamotoi* and remain infected long enough to allow naïve vector ticks to acquire infection via feeding. An indirect method to determine a reservoir species tests both wildlife and ticks in a survey plot for *B. miyamotoi* infection. A direct method live-traps animals and tests both the animal and any ticks on them for *B. miyamotoi*. Xenodiagnostic ticks are then fed on the animal(s) and tested for *B. miyamotoi* to determine transmission capabilities.

It is unclear if animals, particularly reservoir species, remain persistently infected with *B. miyamotoi*. *Bb*sl species are known to cause persistent infections in rodents [117]. A survey of *B. miyamotoi* in rodents and their ticks found that there was no correlation between rodent age, month of infection, and infection presence. Infection rates remained stable across the survey. In the same study, *B. burgdorferi* infection rates increased throughout the survey [118]. Furthermore, while *Bb*sl species have a strong positive association with rodent population density, maintenance of *B. miyamotoi* appears to be independent of rodent population density [119]. This suggests that *B. miyamotoi* and *Bb*sl use different maintenance strategies [69].

*Borrelia miyamotoi* has been found in a variety of mammals, birds, and associated ticks. Discerning how *B. miyamotoi* is maintained in reservoir species, other hosts, and vector ticks is critical for surveillance, risk assessment, and mitigation.

**Table 3 pathogens-12-00267-t003:** *Borrelia miyamotoi* animal infections.

	% Positive	Countries
**Small Mammals**		
Striped field mouse (*Apodemus agrarius*)	3.13% (1/32); 13.2% (7/53); 7.0% (11/157)	Austria [120], Croatia [121], Poland [112]
Small Japanese field mouse (*Apodemus argenteus*)	0.7% (1/137)	Japan [118]
Yellow-necked mouse (*Apodemus flavicollis*) *	0.7% (1/131); 0.9% (1/102) ^a^ & 1.5% (1/67) ^b^; 3.6% (3/84); 2.0% (1/49); 0.4% (1/251) ^c^ & 2.2% (1/46) ^d^	Croatia [121], Hungary [111], Poland [112], Romania [122], Slovenia [123]
Large Japanese field mouse (*Apodemus speciosus*)	2.2% (10/446)	Japan [118]
Wood mouse (*Apodemus sylvaticus*)	14.3% (3/21)	Netherlands [94]
European hedgehog (*Erinaceus europaeus*)	5.0% (3/60)	Czechia [124]
Common vole (*Microtus arvalis*)	12.5% (1/8)	Netherlands [94], Slovakia [59]
Meadow vole (*Microtus pennsylvanicus*)	0.7% (1/146)	Canada [113]
Bank vole (*Myodes glareolus*) *	5.5% (4/72); 8.8% (3/34); 3.1% (1/32)	France [86], Netherlands [94], Romania [122], Switzerland [103]
Grey red-backed vole (*Myodes rufocanus*)	1.0% (2/195)	Japan [118]
Jumping mouse (*Napaeozapus insignis*)	14.3% (3/21)	Canada [113]
Dusky-footed woodrat (*Neotoma fuscipes*)	16.7% (1/6)	United States [125]
Brush mouse (*Peromyscus boylii*)	2.8% (2/71)	United States [125]
California mouse (*Peromyscus californicus*)	16.7% (4/24)	United States [125]
White-footed mouse (*Peromyscus leucopus*) *	6.4% (36/556) ^d^ & 2.3% (2/86) ^e^; 0.5% (3/625)	United States [69], Canada [126]
Deer mouse (*Peromyscus maniculatus*)	2.9% (1/34)	Canada [113]
Eastern grey squirrel (*Sciurus carolinensis*)	25.0% (1/4)	Canada [113]
Red squirrel (*Sciurus vulgaris*)	13.6% (3/22)	Czechia [124], Hungary [127]
Common Shrew (*Sorex araneus*)	16.7% (1/6)	Croatia [121]
Muller’s giant Sunda rat (*Sundamys muelleri*)	33.3% (1/3)	Malaysia [128]
Eastern chipmunk (*Tamias striatus*)	15.4% (2/13)	Canada [126]
**Large mammals**		
Père David Deer (*Elaphurus davidianus*)	2.3% (1/43)	China [57]
**Birds**		
European greenfinch (*Carduelis chloris*)	25% (1/4)	Netherlands [94]
Wild turkey (*Meleagris gallopavo*)	56.0% (35/60)	United States [109]
Great tits (*Parus major*)	50% (1/2)	Netherlands [94]
Ostrich (*Struthio camelus*)	16.7% (1/6)	Czechia [110]

^a^ Positive skin samples, ^b^ Positive spleen samples, ^c^ Retrospective study, ^d^ Prospective study, ^e^ Positive blood samples. * Xenodiagnostic ticks were used to confirm species as a reservoir host.

**Table 4 pathogens-12-00267-t004:** Animal-associated *B. miyamotoi* tick infections.

	Tick	Countries
**Mid-Sized Mammals**		
Beech marten (*Martes foina*)	*I. ricinus*	Belgium, Netherlands [129]
European pine marten (*Martes martes*)	*I. ricinus*	Belgium, Netherlands [129]
European polecat (*Mustela putorius*)	*I. ricinus*	Belgium, Netherlands [129]
**Large Mammals**		
Cattle (*Bos primigenius tarus)*	*I. ricinus*	Germany [130]
Dog (*Canis lupus familiaris*)	*I. ovatus*, *I. hexagonus*, *I. ricinus*, *I. persulcatus*, *Dermacentor reticulatus*	Germany [131,132], Japan [133], Latvia [134], Russia [56]
Goat (*Capra aegagrus hircus*)	*I. ricinus*	Germany [130]
Roe deer (*Capreolus capreolus*)	*I. ricinus*	Germany [132], Poland [135], Spain [136]
Red deer (*Cervus elaphus*)	*I. ricinus*	Poland [137]
Raccoon dog (*Nyctereutes procyonoides*)	*I. ricinus*	Denmark [138]
White-tailed deer (*Odocoileus virginianus*)	*I. scapularis*	United States [114,115,139]
Wild boar (*Sus scrofa*)	*I. ricinus*	Poland [136]
**Birds**		
Northern cardinal (*Cardinalis cardinalis*)	*I. dentatus*	United States [140]
Veery (*Catharus fuscescens*)	*I. scapularis*	Canada [126]
Hermit Thrush (*Catharus guttatus*)	*I. scapularis, I. dentatus*	Canada [126], United States [140]
European robin (*Erithacus rubecula*)	*I. ricinus*	Netherlands [93], Sweden [67]
Song Sparrow (*Melospiza melodia*)	*I. scapularis*	Canada [126]
Common redstart (*Phoenicurus phoeincurus*)	*I. ricinus*	Sweden [67]
Common chiffchaff (*Phylloscopus collybita*)	*I. ricinus*	Netherlands [93]
Eurasian wren (*Troglodytes troglodytes*)	*I. ricinus*	Sweden [67]
Common blackbird (*Turdus merula*)	*I. ricinus*	Moldova [91], Netherlands [93], Poland [135]
American robin (*Turdus migratorius*)	*I. dentatus*	United States [140]
Song thrush (*Turdus philomelos*)	*I. ricinus*	Netherlands [93]

## 8. Transmission

*Borrelia miyamotoi* can be transmitted both horizontally and vertically [141]. Horizontal transmission occurs between a vector and a host or vice versa. Vertical transmission occurs transovarially between a female tick and her progeny. Both modes are suspected to play key roles in persistence of *B. miyamotoi* in the environment. A proposed enzootic transmission cycle depicting both transmission methods and their potential interactions is shown in Figure 1.

Horizontal transmission of *B. miyamotoi* from tick to host appears to have varied success using laboratory models. Mice challenged with *B. miyamotoi* by horizontally infected ticks resulted in low infection prevalence in both mice and xenodiagnostic larvae [64]. Yet, *B. miyamotoi* can be transmitted from infected ticks to naïve mice during the first 24 h of feeding. Within this window of time, only 10% of the mice had detectable *B. miyamotoi* DNA in their blood. If ticks were allowed to feed to repletion, 73% of the mice had detectable *B. miyamotoi* DNA, though no xenodiagnostic ticks were fed on these mice [142]. This likely indicates that *B. miyamotoi* in the tick salivary glands enter the host first. Then, as the tick feeds, spirochetes in the midgut migrate to the salivary glands to contribute to the infection [3,64].

Tick acquisition of *B. miyamotoi* via feeding on an infected immunocompetent host is variable and potentially limited by periods of spirochetemia. Only 1–12% of post-molt ticks were positive for *B. miyamotoi* when fed to repletion on infected reservoir species *P. leucopus* and infected CD-1 mice [64]. This may be due to *B. miyamotoi* colonizing the skin at a low rate combined with brief periods of spirochetemia, or that infection was not sustained through the tick molting process [69]. All naïve ticks fed on infected severe combined immunodeficient (SCID) mice with high, persistent spirochetemia became infected, though infection was frequently lost during the molt [19]. This suggests that spirochetemia may play a role in horizontal transmission from host to tick and that vertical transmission may be more important for *B. miyamotoi* maintenance as horizontal transmission is often inefficient.

*Borrelia miyamotoi* can be vertically transmitted from the female tick to her offspring. The rate of transovarial transmission appears to be positively correlated with the maternal bacterial load. The higher the infection density in the female, the more likely her egg clutches will be infected with higher bacterial loads. Seven out of ten *I. scapularis* females that tested positive for *B. miyamotoi* produced infected larvae, with the filial infection rate of infected clutches being 3.3–100%, median 71%. The three females that did not produce infected clutches had the lowest bacterial loads [64]. Another group saw 10/11 infected *I. scapularis* females produce infected clutches, with infection in progeny being 36–100%. This study also saw similar oviposition rates in *B. miyamotoi*-infected *I. scapularis* females compared with uninfected females, indicating that *B. miyamotoi* infection does not appear to impact tick fecundity [115]. These studies support that vertical transmission is relied on more heavily for *B. miyamotoi* maintenance than horizontal transmission.

Though there is minimal evidence, additional suggested transmission routes of *B. miyamotoi* are vertical transmission in mammals and co-feeding in ticks. A study found a pregnant jumping mouse to have a fully disseminated infection, resulting in *B. miyamotoi* detectable in placental and fetal tissues [113]. It is unclear if the pregnancy had reached full term if the offspring would have been viable or remained infected. Furthermore, it is unclear if the pregnancy led to the disseminated infection or if dissemination was due to unknown factors. Additionally, it has been suggested when both infected and naïve ticks feed simultaneously on the same naïve host, *B. miyamotoi* may be transmitted to the naïve tick, known as co-feeding [64]. Further studies need to be conducted to better understand these potential transmission routes.

As ticks can be infected with *B. miyamotoi* prior to their first blood meal, it is important to note that larval, nymphal, and adult ticks can transmit *B. miyamotoi* to humans. A study by Breuner et al. showed that over half (57%) of CD-1 mice exposed to a single transovarially infected larva had evidence of *B. miyamotoi* infection [143].

## 9. Phylogenetics

The earliest known *B. miyamotoi* isolates were collected in Japan in 1995: isolate FR64 from *A. argenteus* field mice and isolates HT24, HT31, Hk004, and NB103/1 from unfed *I. persulcatus* ticks [4]. In the decades since, numerous additional *B. miyamotoi* isolates have been recovered and described from around the world. Table 5 shows reference genomes and chromosomes for 36 different strains. These strains have been utilized in multifarious research objectives including phylogenetics studies.

*Borrelia miyamotoi* isolates collected from Asia, Europe, and North America have high identity; however, isolates are genetically distinct between continents and can be divided into three geography-based lineages [144]. Isolates within these large geographic lineages often are vectored by more than one tick species and cluster based on vector species. Sequence variation also exists between isolates vectored by the same tick species [144]. The genetic diversity of *B. miyamotoi* isolates can likely be attributed to genetic drift, natural selection, and vector tick speciation [145]. This variability has an unknown impact on population phenotypes [144]. Inferences of isolate infectivity can be made based on *B. miyamotoi* infection data from the different regions, but this has not been a focus of research yet.

## 10. Laboratory Studies on *B. miyamotoi*

### 10.1. Culturing

*Borrelia* species are known for their fastidious nature regarding in vitro cultivation, and *B. miyamotoi* is no exception. The ability to successfully cultivate an organism is at the basis of research and of utmost importance. Indeed, the development of Barbour-Stoener-Kelly (BSK) media was groundbreaking for *Bb*sl research [146]. Unfortunately, *B. miyamotoi* does not grow consistently well in BSK media. Several modified versions have been developed and tested including BSKII, BSK-H, and Modified-Kelly-Pettenkofer (MKP) media, all of which have different composing concentrations, but use rabbit serum for the serum source [147,148,149].

Modified-Kelly-Pettenkofer medium was tested with strains HT31 and M1029. Optimal growth for both strains (maximum growth densities (MGD) 2.5 × 10^7^ and 1 × 10^7^ cells/mL, respectively) occurred in media containing 50% pooled human serum [150]. These strains also grew successfully in a modified version of MKP containing 10% fetal calf serum (MKP-F) [151].

Barbour-Stoener-Kelly-IIB medium was tested with nine North American isolates and three Japanese isolates. Barbour-Stoener-Kelly-IIB media supported all Japanese isolates tested, HT31, HT24, and FR64b (MGD 9.0 × 10^7^, 9.0 × 10^7^, and 8.5 × 10^7^ cells/mL, respectively), but only supported the growth of one North American isolate to a lesser degree, CA17-2241 (MGD 4.2 × 10^6^ cells/mL) [31].

Barbour-Stoener-Kelly-R medium, a diluted version of BSK-II that is supplemented with Lebovitz’s L-15, mouse serum, and fetal calf serum, was found to support the growth of all nine North American isolates tested: RI13-2395, CT13-2396, CT15-0838, CT15-0839, CT15-0840, CT15-0841 (MGD 1.2–1.5 × 10^8^ cells/mL), MN16-2304 (MGD 6.8 × 10^7^ cells/mL), CA17-2241 (MGD 6.2 × 10^7^ cells/mL), and LB-2001 (MGD 3.6 × 10^7^ cells/mL), and all three Japanese isolates tested, HT31 (MGD 7.0 × 10^7^ cells/mL), HT24 (MGD 1.5 × 10^8^ cells/mL), and FR64b (MGD 1.3 × 10^8^ cells/mL). Barbour-Stoener-Kelly-R media also supports the growth of *B. miyamotoi* from blood of infected patients and infected tick homogenate [31]. It has also been shown that *B. miyamotoi* can be isolated from SCID blood if co-cultured in BSK-R medium with *I. scapularis* embryonic cells (ISE6) [31].

### 10.2. Genomics and Pathogenesis

The development of successful *B. miyamotoi* culturing techniques and complete genome sequencing of *B. miyamotoi* has allowed for genetic manipulations of *B. miyamotoi*.

*Borrelia* species have a genomic structure of an approximate 1-megabase chromosome and numerous linear and circular plasmids [152]. A recent study investigated plasmid differences between North American isolates LB-2001 and CT13-2396. Both isolates have the standard chromosome, with plasmid variation: LB-2001 has 12 plasmids, 8 linear and 4 circular, and CT13-2396 has 14 plasmids, 9 linear and 5 circular. Upon further investigation, the differences in the plasmid contents were found to be the result of alternative rearrangements [153]. This revealed that *B. miyamotoi* has the capability for genetic rearrangement among isolates of the same geographic lineage [154]. The same group also investigated the genetic stability of *B. miyamotoi* and found that when passaged over 15 months in vitro, LB-2001 retained plasmids and infectious phenotype [136].

Conversely, a study found that *B. miyamotoi* loses complement binding inhibitory protein (CbiA), an outer surface protein that conveys serum resistance, with prolonged in vitro passage. As CbiA binds classical and alternative pathway complement factors FH, C3, C3b, C4b, and C5, the loss decreases serum resistance [155]. It is unclear if the loss of CbiA can be attributed to plasmid loss, rearrangement, or alternative gene expression. A caveat of these studies is that plasmid loss was investigated in North American strain LB-2001, while CbiA loss was investigated in Japanese strain HT31, which may exhibit differing stability. Furthermore, other *Borrelia* species, especially *Bb*sl, demonstrate reduced genome and plasmid content with extended in vitro passage, often impacting growth and infectivity [156,157,158]. This research suggests that plasmid loss in *B. miyamotoi* may occur.

As genetic manipulation of *B. miyamotoi* is still developing, heterologous expression of *B. miyamotoi* genes in *Bb*sl species has allowed researchers to characterize in vitro effects [155,159,160]. This method has identified multiple novel *B. miyamotoi* proteins that convey serum resistance including fibronectin binding proteins FbpA and FbpB, BOM1093, Vlp-15/16, and Vlp18. Fibronectin binding proteins A and B were identified in strains FR64b and LB-2001 with 85% and 95% similarity, respectively. These proteins are orthologs of *Bb*sl protein BBK32 and inhibit the classical complement pathway by binding factor C1r [161]. Fibronectin binding protein A binds human fibronectin and is capable of restoring serum resistance in a serum-sensitive *B. burgdorferi* strain, while FbpB does not [161]. Though the role of FbpA in fibronectin binding is unknown, FbpA may be a virulence determinant, as fibronectin binding often mediates *Borrelia* dissemination and colonization [162]. BOM1093 was identified in strain MYK3 and is a vitronectin-binding protein that confers serum resistance through the inhibition of complement C5b7 complex formation and C9 polymerization. The capability of BOM1093 to bind vitronectin is suspected to convey additional unknown serum resistance mechanisms [163]. Variable major proteins Vlp-15/16 and Vlp18 were identified in strain HT31 to inhibit classical and alternative pathway complement. Additionally, Vlp-15/16 binds plasminogen in an FH-independent manner, which potentially plays a role in tissue dissemination [17]. All of these proteins exhibiting redundancy in complement inhibition and serum resistance likely play an important role in innate immune system evasion.

## 11. Conclusions

While *B. miyamotoi* research has rapidly expanded from its discovery in 1995, its pathogenicity in 2011, to the current classification of BMD as an emerging infectious disease, much remains unknown. The host–pathogen interactions that result in BMD, with symptoms ranging from sub-clinical to severe, are mostly speculative. Little is known of tissue colonization, dissemination, immune evasion, and the related mechanisms in ticks, animal hosts, and human patients. While herein, we proposed a *B. miyamotoi* enzootic transmission cycle, this is based on inferences and has not been explicitly analyzed. *Borrelia miyamotoi* maintenance in nature appears to rely on both horizontal and vertical transmission; however, the extent to which these processes occur is unclear.

It can be concluded that there is a great need for increased awareness of *B. miyamotoi* in the public and in healthcare, and a standardized diagnostic method for *B. miyamotoi* infection needs to be developed. *Borrelia miyamotoi* research still has a great breadth for expansion.

## Figures and Tables

**Figure 1 pathogens-12-00267-f001:**
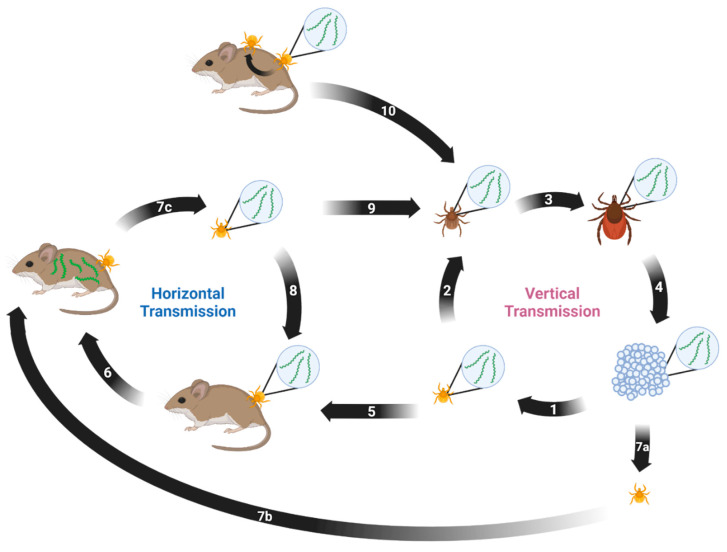
Proposed enzootic transmission cycle for *Borrelia miyamotoi.* (1) Larvae hatch from eggs laid by a *B. miyamotoi*-infected female *Ixodes* tick. Transovarial transmission occurs, resulting in partial larval infection. (2) The infected larva takes first blood meal and molts, remaining infected. (3) The infected nymph takes second blood meal and molts, remaining infected. (4) The infected adult female takes third blood meal, mates, and lays eggs. (5) The infected tick feeds on naïve host. (6) The host acquires *B. miyamotoi* from the infected tick. (7) (a) The naïve larva (b) feeds on the infected host and (c) acquires *B. miyamotoi* via blood meal. (8) The infected tick feeds on the naïve host. (9) The infected larva takes blood meal and molts into a nymph, remaining infected. (10) The naïve tick acquires *B. miyamotoi* via co-feeding near an infected tick. Created with BioRender.com (accessed on 9 January 2023).

**Table 5 pathogens-12-00267-t005:** *Borrelia miyamotoi* strains with reference genome or chromosome sequences.

Strain	Origin	Source	Accession
LB-2001	North America	*I. scapularis*	CP006647
CT13-2396	North America	*I. scapularis*	NZ_CP017126
C14D4	North America	Human, blood	NZ_CP010308
CA17-2241	North America	*I. pacificus*	NZ_CP021872
FR64b	Japan	*A. argenteus*, blood	NZ_CP004217
HT31	Japan	*I. persulcatus*	NZ_AP024371
HT24	Japan	*I. persulcatus*	NZ_AP024372
Hk004	Japan	*I. persulcatus*	NZ_AP024373
NB103/1	Japan	*A. argenteus*, blood	NZ_AP024374
MYK1	Japan	*I. pavlovskyi*	NZ_AP024375
MYK2	Japan	*I. persulcatus*	NZ_AP024391
MYK3	Japan	*I. persulcatus*	NZ_AP024392
MYK4	Japan	*I. persulcatus*	NZ_AP024393
MYK5	Japan	*I. persulcatus*	NZ_AP024394
Y14T1	Japan	*I. persulcatus*	NZ_AP024398
Y15T1	Japan	*I. persulcatus*	NZ_AP024399
Y14T18	Japan	*I. ovatus*	NZ_AP024400
Yekat-1	Russia	Human, plasma	NZ_CP024333
Yekat-6	Russia	Human, plasma	NZ_CP024316
Yekat-17	Russia	Human, plasma	NZ_CP037215
Yekat-18	Russia	Human, plasma	CP037471
Yekat-19	Russia	Human, plasma	NZ_CP037058
Yekat-21	Russia	Human, plasma	NZ_CP036914
Yekat-31	Russia	Human, plasma	NZ_CP036726
Yekat-76	Russia	Human, plasma	NZ_CP036557
Izh-4	Russia	Human, plasma	NZ_CP024390
Izh-5	Russia	Human, plasma	NZ_CP024205
Izh-14	Russia	Human, plasma	CP024371
Izh-16	Russia	Human, plasma	CP024351
NL-IR-1	Europe	*I. ricinus*, eggs	NZ_CP044783
NL-IR-2	Europe	*I. ricinus*, eggs	NZ_CP044625
CZ-F1E	Europe	*I. ricinus*, eggs	NZ_CP046389
CZ-F190E	Europe	*I. ricinus*, eggs	NZ_CP046388
M12C4	Mongolia	*I. persulcatus*	NZ_AP024395
M15A8	Mongolia	*I. persulcatus*	NZ_AP024396
M20E6	Mongolia	*I. persulcatus*	NZ_AP024397

Accession numbers taken from GenBank.

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
