# Peer review of "Borrelia miyamotoi: A Comprehensive Review"

_pathogens, 2023, doi:10.3390/pathogens12020267_

Round 1
Reviewer 1 Report
P2, line 49: Researchers once hypothesized that STARI was caused by the spirochete, Borrelia lonestari; however, further research did not support this idea. https://doi.org/10.1086/427289
P2, line 67: globally is not correct, not in Africa, Australia, and South America.
P2, Line 86: Most patients can clear a B. miyamotoi infection without medical intervention due to the adaptive immune response. Reference?
P4, Line 149: “GlpQ allows for distinction between B. miyamotoi and Bbsl, but it does not distinguish between B. miyamotoi and other RF species”. Here the authors misunderstood the cited paper. The authors are talking about PCR – DNA sequences, not a protein. They are several B. miyamotoi PCR assays using glpq gene – it can easily distinguish between B. miyamotoi and other RF species. For example, the glpq Identities between B. miyamotoi and B. hermsii is 0.84.
P4, Serodiagnosis part: The authors may need to review more papers. In Krause et al, 2015 review: “B. miyamotoi GlpQ protein is no more than 50% identical to the GlpQ proteins of some other bacterial pathogens, such as Klebsiella pneumoniae and Salmonella enterica. On the basis of this distance, B. miyamotoi GlpQ is not expected to cross-react with anti-GlpQ antibodies elicited by other disease agents, but this has not been established as yet.”
P4, Line 225-230: The authors should do a comprehensive review of the cases. For example, 43 cases in France https://doi.org/10.3389/fmed.2020.00055
P6, Line 249: B. miyamotoi DNA was also detected in I. ovatus, I. pavlovskyi, I. nipponensis, and Haemaphysalis concinna
P6, Line283: “As Bbsl and B. miyamotoi colonize the same tick vector, it is likely that B. miyamotoi use similar mechanisms”. Bbsl uses ospA to bind TROSPA in vector ticks. Where is the evidence/reference of a “similar mechanisms” in B. miyamotoi?
P7, table1: where is I. pacificus?
P8, Animal Infections: It is worth to cite that white-tailed deer permits B. miyamotoi to remain viable in the feeding deer ticks, in marked contrast to the situation with the Lyme disease agent.
P15, Culturing: this part may need to merge with 4.1. Microscopy. Culturing and Microscopy
Author Response
Responses to Reviewer 1
P2, line 49: Researchers once hypothesized that STARI was caused by the spirochete, Borrelia lonestari; however, further research did not support this idea. https://doi.org/10.1086/427289
-P2, line 47: We originally included B. lonestari simply as it groups with RF, but is vectored by a hard tick, however as we edited, we did not notice that it did not read as such. We removed the reference to B. lonestari
P2, line 67: globally is not correct, not in Africa, Australia, and South America.
-P2, line 68: We replaced the word “globally” with widely. We made similar changes throughout the manuscript.
P2, Line 86: Most patients can clear a B. miyamotoi infection without medical intervention due to the adaptive immune response. Reference?
-P2, line 90: This was initially a speculation on our (the authors) part. We rephrased this to include data from mouse models showing that immunocompetent animals are able to clear the infection. We also adjusted our phrasing to make it clearer that this is only a speculation in humans.
P4, Line 149: “GlpQ allows for distinction between B. miyamotoi and Bbsl, but it does not distinguish between B. miyamotoi and other RF species”. Here the authors misunderstood the cited paper. The authors are talking about PCR – DNA sequences, not a protein. They are several B. miyamotoi PCR assays using glpQ gene – it can easily distinguish between B. miyamotoi and other RF species. For example, the glpQ Identities between B. miyamotoi and B. hermsii is 0.84.
-P4, line 150: We removed this sentence, thank you for the clarification.
P4, Serodiagnosis part: The authors may need to review more papers. In Krause et al, 2015 review: “B. miyamotoi GlpQ protein is no more than 50% identical to the GlpQ proteins of some other bacterial pathogens, such as Klebsiella pneumoniae and Salmonella enterica. On the basis of this distance, B. miyamotoi GlpQ is not expected to cross-react with anti-GlpQ antibodies elicited by other disease agents, but this has not been established as yet.”
-P4, line 184: We removed this sentence and replaced it with a more generalized sentence on the potential for cross-reactivity that reflects Krause et al 2015 properly. Thank you again.
-We rearranged the order of paragraphs in the serodiagnosis section. The section now starts with explaining serodiagnosis, instead of the paragraph on GlpQ, and we moved the section on GlpQ (lines 180-185) down.
P4, Line 225-230: The authors should do a comprehensive review of the cases. For example, 43 cases in France https://doi.org/10.3389/fmed.2020.00055
-P5, line 229-230: The authors initially did not include the 43 cases in France as there was some debate about the validity over the initial study. However, we now included these cases along with a couple additional cases from France.
P6, Line 249: B. miyamotoi DNA was also detected in I. ovatus, I. pavlovskyi, I. nipponensis, and Haemaphysalis concinna
-P5, line 249-254: We adjusted the sentences to include these species.
P6, Line283: “As Bbsl and B. miyamotoi colonize the same tick vector, it is likely that B. miyamotoi use similar mechanisms”. Bbsl uses ospA to bind TROSPA in vector ticks. Where is the evidence/reference of a “similar mechanisms” in B. miyamotoi?
-P6, line 290: This was speculative. We changed this sentence to now read that it is unknown if B. miyamotoi uses similar mechanisms as Bbsl.
P7, table1: where is I. pacificus?
-P7, Table 1: Thank you for noticing this, we now included I. pacificus.
P8, Animal Infections: It is worth to cite that white-tailed deer permits B. miyamotoi to remain viable in the feeding deer ticks, in marked contrast to the situation with the Lyme disease agent.
-P8, line 331-337: We strongly agree that it is worth to cite white-tailed deer as a potential reservoir species. A new paragraph dedicated to white-tailed deer has been added.
P15, Culturing: this part may need to merge with 4.1. Microscopy. Culturing and Microscopy
-The authors agree that sections 9 and 10 needed to be adjusted. We do not think that the section on culturing belongs with microscopy as the data on culturing is in the context of laboratory research and the microscopy section is looking at human diagnosis. Instead, Section 9 has been truncated to Phylogenetics only. Section 10 “Laboratory Studies on B. miyamotoi”, now has the subsections of “Culturing” and “Genomics and Pathogenesis”. We think this is better suited as culturing is instrumental for laboratory studies, so research on media and culturing leads into genomic and pathogenesis work nicely.
-The authors would like to thank Reviewer 1 again for their feedback. All comments were highly constructive and have resulted in a better manuscript.
Reviewer 2 Report
This literature review focuses on Borrelia miyamotoi, the cause of the newly emerging tick-borne disease. The review is solid and is sure to be welcomed by readers. I have only a few minor comments that should be taken into consideration before publishing the paper:
1. Key words: because Borrelia miyamotoi is included in the title, it is better to use here "Borrelia miyamotoi disease (BMD)" instead of the bacterium species.
2. Key words: Ixodes should be in italics.
3. L24: Borrelia should be in italics.
4. L38, 56, 158, and everywhere else in the text where it applies: Do not start the sentence with abbreviation.
5. L48: “Argasid” should be lowercase letter (only original Latin taxon names with a capital).
6. L83, 143, and everywhere else in the text where it applies: please give the full name of the proteins/genes/mutant mouse line/etc. when they first appear (and the abbreviation in parentheses).
7. L95: “i.e.,” instead of “ie” and please give the full name of the method).
8. L159: In this case, GlpQ refers to a gene, so it should be in italics.
9. L264: 0% prevalence means no infection, so it's better to write it directly, e.g., “provinces in which infection has not been reported”.
10. Table 3 would be more convenient to read if countries and references were in the last column and prevalence data in the second one.
11. Table 4: remark as above.
12. L397: “transmission routes” would be better than “transmission methods”.
13. L417: “Reference genomes are available” should be “Reference complete genomes or chromosomes are available” (Not all of the 36 strains on the list have their entire genomes sequenced, including plasmids).
14. Table 5: I suggest a change in the title, e.g., “Borrelia miyamotoi strains with reference genome or chromosome sequences”.
15. Table 5: please add lacking references or remove this column (these papers have already been cited in the body text).
16. L476: add reference to BSK medium.
17. L481, 485, 489 – should be “medium”, not “media”.
18. L497 – add additional information to this sentence or move it at the end of the previous paragraph.
Author Response
Responses to Reviewer 2
- Key words: because Borrelia miyamotoi is included in the title, it is better to use here "Borrelia miyamotoi disease (BMD)" instead of the bacterium species.
-We have made this change.
- Key words: Ixodes should be in italics.
- We have corrected this.
- L24: Borrelia should be in italics.
- We have corrected this.
- L38, 56, 158, and everywhere else in the text where it applies: Do not start the sentence with abbreviation.
- We have corrected this throughout the manuscript.
- L48: “Argasid” should be lowercase letter (only original Latin taxon names with a capital).
- We have corrected this.
- L83, 143, and everywhere else in the text where it applies: please give the full name of the proteins/genes/mutant mouse line/etc. when they first appear (and the abbreviation in parentheses).
-Thank you for noticing this, it has been corrected.
- L95: “i.e.,” instead of “ie” and please give the full name of the method).
-line 99: We removed the mention of the method as we did not think it benefited the manuscript.
- L159: In this case, GlpQ refers to a gene, so it should be in italics.
-The authors ultimately rearranged the serodiagnosis section and in the process the authors removed that sentence. We did take a closer look at all other references to GlpQ to make sure the correct emphasis was used.
- L264: 0% prevalence means no infection, so it's better to write it directly, e.g., “provinces in which infection has not been reported”.
-line 268: The authors agree, and we have changed this.
- Table 3 would be more convenient to read if countries and references were in the last column and prevalence data in the second one.
- We have made the suggested change.
- Table 4: remark as above.
- We have made the suggested change.
- L397: “transmission routes” would be better than “transmission methods”.
-line 412: We have corrected this.
- L417: “Reference genomes are available” should be “Reference complete genomes or chromosomes are available” (Not all of the 36 strains on the list have their entire genomes sequenced, including plasmids).
-Line 432: Thank you for noticing this. We changed the sentence to accurately reflect this.
- Table 5: I suggest a change in the title, e.g., “Borrelia miyamotoi strains with reference genome or chromosome sequences”.
-line 445: Thank you again for catching this, the table name has been changed.
- Table 5: please add lacking references or remove this column (these papers have already been cited in the body text).
-The authors removed the reference column for this table as not all strains have publications and because the existing publications have all been cited in text. The authors agree with Reviewer 2 that listing accession numbers from GenBank is sufficient for this table.
- L476: add reference to BSK medium.
- We have corrected this.
- L481, 485, 489 – should be “medium”, not “media”.
-The authors believe we have corrected all of the medium/media.
- L497 – add additional information to this sentence or move it at the end of the previous paragraph.
-line 475: The authors rearranged sections 9 and 10, and in this adjustment, we added more information on culturing B. miyamotoi with ISE6 cells and moved this sentence to the end of the previous paragraph.
-The authors would like to kindly thank Review 2 again for catching many errors, allowing us to put forth a more professional manuscript.
Round 2
Reviewer 1 Report
Accept in present form
Author Response
23 January 2023
Dear MDPI Editorial Staff,
Re: manuscript reference number pathogens-2154165
Please find attached a revised version of our manuscript.
The authors would like to graciously thank the editors for taking the time to review our manuscript.
- Line 2 – regarding the title, did the authors forget to include the word “fever” after “Relapsing”?
- We have changed the title.
- Line 8 (also on line 46) – re-spell “hemisphere” to “Hemisphere”.
- We have corrected this.
- Line 139 – re-word the end of the sentence as follows: “specialized media, and organisms can then be readily visualized”.
- We have made this edit. We chose to leave out “readily” as this method is not used frequently, so we don’t yet know if the technique is reliable.
- Line 181 – change “to” to “into”.
- We have corrected this.
- Lines 186-188 – part of this sentence can be improved by rewording it as follows: “….using conventional assay systems such as an ELISA and a Western blot [46,47]”.
- We have corrected this.
- Line 191 – the word “be” should be inserted after “to”.
- This has been added.
- Line 218 – after “Patients”, change “that” to “who”.
- We have corrected this.
- Line 288 – after “burgdorferi”, insert “organisms”.
- We corrected this, and chose to use “species” instead.
We hope that these revisions are sufficient to make our manuscript suitable for publication in MDPI Pathogens and look forward to hearing from you at your earliest convenience.